# Ongoing Community-Based Whole-Food, Plant-Based Lifestyle Effectively Preserves Muscle Mass during Body Mass Loss

**Boštjan Jakše** [1,*] **, Barbara Jakše** [1] **, Uroš Godnov** [2] **and Stanislav Pinter** [3]

1 Independent Researcher, 1230 Domžale, Slovenia; barbara.tursic@gmail.com
2 Department of Computer Science, Faculty of Mathematics, Natural Sciences and Information Technologies, University of Primorska, 6000 Koper, Slovenia; uros.godnov@ikp-dqi.com
3 Basics of Movements in Sport, Faculty of Sport, University of Ljubljana, 1000 Ljubljana, Slovenia; stanislav.pinter@fsp.uni-lj.si
* Correspondence: bj7899@student.uni-lj.si; Tel.: +386-(4)-1278586

**Abstract:** Body fat and muscle mass showed opposing associations with mortality. The results of research on the effectiveness of popular body mass (BM) loss diets in obese subjects showed 20 to 30% loss of muscle mass within the total BM loss; conversely, when the subjects used a whole-food, plant-based (WFPB) diet, the loss was up to 42%. Therefore, we suggest an improvement. The aim of this retrospective analysis of data was to examine the assessment of changes in the body composition of 217 participants from all over Slovenia who joined our ongoing, community-based WFPB lifestyle programme from 2016 to 2021 and underwent two successive measurements of medically approved bioelectrical impedance. The WFPB lifestyle programme consisted of (i) nutrition, (ii) physical activity (PA) and (iii) a support system. The primary outcomes included the (vector of) change of body fat mass (BFM) per body height (BH), fat-free mass (FFM) per BH and whole-body phase angle (PhA) from the initial values to the first follow-up (FU) of the whole sample and for both sexes. Further, we examined the FFM change within the total BM loss according to their BMI classification and depending on how much BM they lost (5 kg < BM ≥ 5 kg) within the FU time (103.6 ± 89.8 day). Participants experienced a decrease in BFM per BH ($-0.02 \pm 0.02$ kg/cm, $p < 0.001$), no change in FFM and an increase in PhA ($0.2 \pm 0.7°$, $p < 0.001$). Importantly, the participants in the obesity BMI class achieved only partial FFM preservation ($-1.5 \pm 3.6$ kg, $p = 0.032$ of FFM loss (20%) within $-7.5 \pm 6.1$ kg, $p < 0.001$ of BM loss). However, the participants who lost BM < 5 kg had a significantly increased FFM ($0.8 \pm 3.2$ kg, $p = 0.001$ of FFM (57%) within $-1.4 \pm 1.8$ kg, $p < 0.001$ of BM loss), whereas the participants who lost BM ≥ 5 kg experienced a decrease in FFM ($-2.2 \pm 3.9$ kg, $p < 0.001$ of FFM ($-25\%$) within $-8.8 \pm 5.2$ kg, $p < 0.001$ of BM loss). To conclude, the WFPB lifestyle, on average, effectively preserved FFM during BM and/or BFM loss among the normal and pre-obesity BMI classes but only partially among the obese BMI class and those who lost ≥ 5 kg of BM. Importantly, a customized PA strategy is needed for obese BMI class participants, where general resistance training is not possible or safe in order to preserve their muscle mass more effectively. In addition, muscle mass preservation is important for further improvements of BM, body composition and visual body image.

**Keywords:** body composition; whole-food; plant-based diet; physical activity; resistance workout; support system; body mass; body fat; muscle mass; phase angle

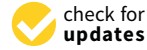



## 1. Introduction

Overweight and obesity are major public health challenges [1,2]. In fact, there are over 50 obesity-related comorbidities that mediate this global burden [3]. A variety of body mass (BM) loss practices are being applied to try to manage overweight and obesity [4–8]. Although the results of previous studies are not conclusive, most of them do not confirm that BM loss can be successfully maintained long-term by using various popular

diets [8–11]. Importantly, physical activity (PA) also has a beneficial effect on various aspects of health [12–14]; however, an appropriate BM can be more effectively controlled with a combination of PA and dietary changes [15–20].

Importantly, because most intervention studies examining the effect of a vegan diet on various health issues and BM loss did not measure body composition (BC) [21–24], it is mostly unclear what the proportions of muscle mass and body fat mass (BFM) were within the total BM loss. Diet-induced BM loss is often accompanied by fat-free mass (FFM) loss, which varies based on the type of BM loss dietary intervention. In brief, the results of research on the effectiveness of popular BM loss diets in obese subjects showed 20 to 30% loss of lean muscle mass within the total BM loss [17,25,26], whereas in some recent studies using a WFPB diet only for overweight or obese participants, the loss was up to 42% [27–30]. These disturbing results may be due to the failure to include an appropriate form and frequency of PA or energy restriction of the implemented dietary plan regardless of whether the subjects were restricted or allowed to eat each meal ad libitum. Nevertheless, as suggested by several researchers, the strategies for reducing muscle mass loss in the BM loss process vary, ranging from (i) including physical activity (e.g., especially resistance workouts) to (ii) introducing various dietary interventions (e.g., increasing their protein intake) [17,19,31,32]. The loss of excess body fat while maintaining the muscle mass are key to the process of further losing excess BM and to sustaining the lost BM [17,31–34] and appetite control in terms of avoiding BM regain [26,35]. Therefore, BC provides important prognostic information on an individual's BM management and mortality risk that are not provided by traditional proxies of adiposity, such as body mass index (BMI) [36–38]. This means that chronic disease and the global obesity epidemic may be underestimated if we only use BMI as a central marker of obesity [38,39].

In addition, bioelectrical impedance is a relatively simple, inexpensive, non-invasive and quick technique to measure BC (by sending a weak electrical current) and is therefore suitable in field studies and larger surveys [40]. Moreover, measuring BC with bioelectrical impedance (BIA) can also provide an estimate of the whole-body phase angle (PhA) parameter, which is a clinically important prognostic marker of physical health [41]. In brief, PhA is a linear method of measuring the relationship between electric electrical resistance (i.e., opposition caused by the substance to the flow of the current) and reactance (i.e., the applied voltage current), expressed as an angle [42]. Simply put, a higher PhA suggests a healthier whole-body condition, whereas a lower PhA indicates a state of worse health [43] and was found to be associated with earlier mortality [44]. However, age, sex, BMI and BC (i.e., extracellular water (ECW)–intracellular water (ICW) ratio, FFM and BFM) are major PA determinants in healthy subjects [41] that are quickly mitigated by nutritional [45] and PA changes [46,47].

There is a sparsity of information on whether a WFPB lifestyle that also includes a PA component (e.g., resistance workouts) could be a method that can effectively mitigate the adverse effect of a WFPB diet alone on muscle mass preservation as a component of BC status. Given the globally growing trend to encourage people to rely more on plant-based diets [48–50], more information on how to preserve muscle mass during BM loss is urgently needed. Therefore, an open research problem is to determine an effective and viable way to improve BC (i.e., losing excess body fat while preserving muscle mass as much as possible) during BM loss.

The aim of this retrospective analysis of data was to examine assessment of changes in the BC of 217 participants (i.e., regardless of their baseline BMI) from all over Slovenia who joined our ongoing, community-based WFPB lifestyle programme. The decision was based on favourable findings in our previous two interventional and one cross-sectional study [51–53] and our favourable clinical findings over the past 12 years using an ongoing, community-based WFPB lifestyle programme in regard to muscle mass preservation during BM loss and less favourable findings in recent studies using only a WFPB diet.

## 2. Materials and Methods

### 2.1. Study Design and Eligibility

This retrospective study protocol was reviewed and approved on 12 April 2021, by the Ethical Committee in the field of sports in Slovenia (approval document no. 033–7/2021–2), and the trial was registered at https://clinicaltrials.gov (accessed on on 29 April 2021) with number NCT04849689 [54]. We analysed two successive BC records of each participant from a wide geographic area of Slovenia who joined our ongoing, community-based WFPB lifestyle programme and were measured by bioelectrical impedance from 2016 to 2021 (all participants joined as a recommendation and were measured in the capital, Ljubljana). The WFPB diet was self-financed, while the support system was mostly free of charge or had only a small registration fee (i.e., for cooking classes or workshops for those who opted to participate). Due to the specific study design (i.e., anonymously retrospectively analysed data to avoid any possible impact of research involvement), informed consent to participate in the study was not required.

We investigated the possible FFM preservation of participants in our ongoing, community-based (free-living) WFPB lifestyle programme during BM loss. The exported and used data were statistically analysed by an independent researcher (a professor of statistics, the co-author). The flow of the analysed data is shown in Figure 1.

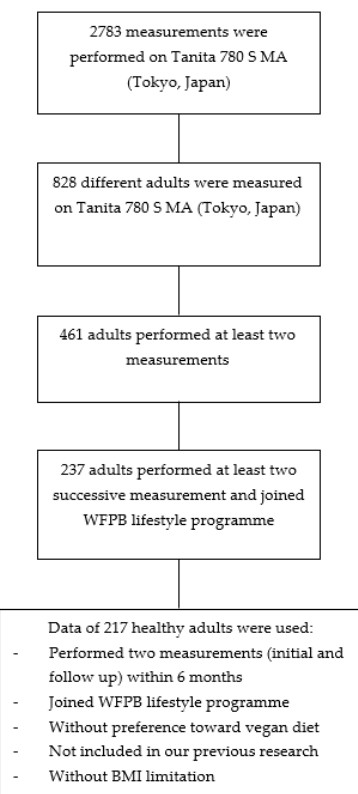

**Figure 1.** The flow of the analysed data.

### 2.2. Subjects

We included data from 217 healthy adult participants from a large geographical part of Slovenia, with a mean age of $41.1 \pm 12.7$ years. The participants whose data we used were on the WFPB lifestyle programme and were measured with a BC analyser at least two successive times at different intervals (i.e., before the start of the programme (initial) and as part of the follow-up (the first follow-up (FU) within 6 months). Other inclusion criteria were as follows: being without a preference toward vegan diets and being older than 18 years before the start of the programme. Furthermore, by "healthy adults", we mean that the data used were from participants without medication use and without serious comorbidities (i.e., having autoimmune or neurodegenerative disorders),

physical limitations and no use of medications (e.g., to lower blood pressure, blood sugar, cholesterol or thyroid disease). Importantly, initial BMI status was not an exclusion criterion. In addition, in this study, we included data only from those subjects within the inclusion criteria who had not participated in our previous research; therefore, there was no present knowledge effect on the results of being included in the study.

### 2.3. Outcomes

The variables in this study included the detailed demographic characteristics of the participants, their anthropometrics, BC indices at two measurements (initial and FU), and associations between their BC change (BMI, BFM and FFM) and PhA, and their baseline Western-type lifestyle and the intervention WFPB lifestyle programme characteristics.

#### 2.3.1. Demographic Characteristics of the Subjects

The baseline demographic characteristics of the study subjects included age, sex, body height, lifestyle (e.g., PA, medication use, alcohol intake and smoking habits), education status and living region.

#### 2.3.2. Anthropometrics and BC Parameters

The primary outcome was the vector of change of BMI, BFM per body height (BH), FFM per BH, and PhA from the initial values to the first FU for the whole sample and by sex. In addition, we examined the FFM change within the total BM loss according to the BMI classification by the World Health Organization (WHO) [55] and depending on how much BM they had lost (5 kg < BM $\geq$ 5 kg). Last, with repeated measures correlation (rmcorr), we estimated the associations between BMI, BFM and FFM, and PhA status.

The BC measures included BH (measured via a standardized medical approved professional personal floor scale with a stand, the Kern MPE 250K100HM (Kern and Sohn, Balingen, Germany) [56], BM, BFM, FFM, total body water (TBM), extracellular water (ECW), intracellular water (ICW), ECW–ICW ratio and PhA, all measured via the same medically approved and calibrated bioelectrical impedance Tanita 780 S MA (Tanita Corporation, Tokyo, Japan) [57] using the corresponding protocols (i.e., BMI was calculated as BM in kilograms divided by the square of height in metres). All anthropometric and BC indices were evaluated by the same experienced investigator (first author) and were conducted according to the manufacturer's recommendation for each instrument [56,57]. In addition, the results from an 8-electrode BC analyser compared with dual-energy X-ray absorptiometry provided accurate measurements of total BFM and FFM in healthy young males and females, regardless of their level of habitual PA [58].

#### 2.3.3. Baseline Western-Type Lifestyle and the Intervention WFPB Lifestyle Programme Characteristics

Our participants, based on an author-developed extensive questionnaire, were previously on average engaged in a Western-type lifestyle, where most meals were composed of animal-based foods (i.e., dairy, various meats, eggs and mostly canned fish) and refined plant-based foods (i.e., bread, pasta, vegetable oils and ultra-processed pastry) with limited amounts of whole-food fruits and vegetables. Furthermore, these data indicated that our participants were in general without a preference toward a WFPB lifestyle. In addition, the analysed PA of the participants showed that their PA was largely habitual, nonorganized or inconsistent (see Section 3.1).

The WFPB lifestyle programme included (1) plant-based food (energy intake: $\geq$90% of whole-food, plant-based diet and $\leq$10% of energy intake from plant-based meal replacements and dietary supplements (e.g., a vitamin $B_{12}$ supplement throughout the year and a vitamin D supplement from October to April and optionally omega-3 long-chain polyunsaturated fatty acids (eicosapentaenoic acid (EPA) and docosahexaenoic acid (DHA)), (2) physical activity (habitual; organized for 2–3 times weekly for at least 30 min at a time of moderate-intensity resistance workouts, and we encouraged them to also perform at least

30 min of low-intensity aerobic activity (i.e., brisk walking) most weekdays and one longer (45–120 min of low- to moderate-intensity aerobic activity (i.e., brisk walking or hiking)) per week), and (3) a support system (workshops and individual nutrition counselling, grocery shopping advice, cooking performance measurements, regular physical activity, regular monitoring, regular meetings, assisting in regular activity and medical support). Of importance, despite the fact that BM management was a core part of our programme, there was no need for calorie counting, and recommended WFPB diet foods were consumed ad libitum to full satiety at each meal. Detailed nutritional information about the prescribed content of our ongoing WFPB lifestyle is provided in a separate article [59].

### 2.4. Statistical Analysis

Statistical analysis was performed using R 4.0.3 with the dplyr [60], ggplot2 [61], arsenal [62] and rmcorr [63] packages. Dplyr was used for data transformation, ggplot2 for data visualization, arsenal for statistical calculation and rmcorr for repeated measures correlation. For numerical variables, we used t-tests for independent samples and paired *t*-tests, whereas for associations between BMI, BFM and FFM, and PhA, a rmcorr test for repeated measures was used [64]. The threshold for statistical significance was $p < 0.05$. For an 84% power of *t*-test to detect significant difference, we needed 111 data pairs. There were no missing data. Data are presented as the means (standard deviation).

## 3. Results

### 3.1. Demographic Characteristics of the Subjects

The basic characteristics of the participants are presented in Table 1. Data from 217 participants were included in the analysis: 145 (66.8%) women and 72 (33.2%) men with an average age of 41.1 ± 12.7 years. Baseline results are given for the whole sample. Importantly, 92 (42%) of participants at baseline did not include PA in their lifestyle. Alcohol intake and smoking were present only in a minority of the sample (7% and 5.1%). Furthermore, regarding the level of education, the results were polarized, which means 103 (47%) finished high school while 114 (52.5%) finished college or more. In addition, the participants were from a wide geographical part of Slovenia, but most of them were from the capital city (26.3%), central (26.7%; excluding the capital city that lies in central Slovenia) and eastern Slovenia (24.4%).

**Table 1.** Baseline characteristics of the participants (mean ± SD, *n* (%)).

| Parameter | N = 217 |
|:---:|:---:|
| Age (years) | 41.1 ± 12.7 |
| Woman (*n* (%)) | 145 (66.8) |
| Man (*n* (%)) | 72 (33.2) |
| BH (cm) | 170.8 ± 8.6 |
| BH of woman (cm) | 166.2 ± 5.6 |
| BH of man (cm) | 180.0 ± 5.9 |
| Physical activity (*n* (%)) [†] | |
| Resistance workout [††] | 48 (22) |
| Walking [‡] | 89 (41) |
| No PA | 92 (42) |
| Alcohol intake (≥3 times/w, *n* (%)) | 14 (7) |
| Smoking habits (yes, *n* (%)) | 11 (5.1) |
| Education status (*n* (%)) | |
| High school | 103 (47.5) |
| College or more | 114 (52.5) |

BH: body height. PA: physical activity. [†] Variables with multiple possible answers. [††] At least 2 workouts per week for 30 min. [‡] At least 3 times per week for 45 min.

### 3.2. Anthropometrics and BC Indices

For the whole sample, for women and men, a significant change was measured from baseline to the FU time (the average FU time was $103.6 \pm 89.8$ days) in terms of a lower BM (BMI), BFM and PhA, whereas FFM was not altered. In terms of sex differences, both women and men exhibited significant improvements in BC status, where both groups had a preserved FFM. Importantly, men started on average in the overweight BMI class, while women were in the normal BMI class [55]. The complete anthropometrics and BC indices for woman and men are presented in Table 2, whereas the whole sample is presented in Table S1.

**Table 2.** Anthropometrics and BC indices (mean $\pm$ SD) by sex.

| Parameter | Woman ($N = 145$) | | | | Man ($N = 72$) | | | |
|---|---|---|---|---|---|---|---|---|
| | Baseline | FU | Mean Diff. | *p*-Value | Baseline | FU | Mean Diff. | *p*-Value |
| BM (kg) | $67.8 \pm 11.8$ | $65.4 \pm 10.4$ | $-2.4 \pm 3.1$ | **<0.001** | $88.0 \pm 17.5$ | $84.2 \pm 15.4$ | $-3.9 \pm 5.6$ | **<0.001** |
| BMI (kg/m$^2$) | $24.5 \pm 4.0$ | $23.6 \pm 3.4$ | $-0.9 \pm 1.1$ | **<0.001** | $27.1 \pm 5.0$ | $25.9 \pm 4.3$ | $-1.2 \pm 1.7$ | **<0.001** |
| BFM/BH (kg/cm) [†] | $0.12 \pm 0.05$ | $0.10 \pm 0.04$ | $-0.02 \pm 0.01$ | **<0.001** | $0.11 \pm 0.06$ | $0.08 \pm 0.05$ | $-0.03 \pm 0.02$ | **<0.001** |
| FFM/BH (kg/cm) [†] | $0.29 \pm 0.04$ | $0.30 \pm 0.04$ | $0.01 \pm 0.02$ | 0.063 | $0.37 \pm 0.05$ | $0.37 \pm 0.05$ | 0 | N/A |
| TBW (kg) | $35.8 \pm 6.3$ | $35.8 \pm 5.4$ | 0 | N/A | $46.7 \pm 8.5$ | $47.7 \pm 7.6$ | $1.0 \pm 7.8$ | 0.261 |
| ECW (kg) | $15.3 \pm 2.6$ | $15.2 \pm 2.3$ | $-0.1 \pm 2.7$ | 0.520 | $18.9 \pm 3.1$ | $18.7 \pm 2.8$ | $-0.2 \pm 3.3$ | 0.587 |
| ICW (kg) | $20.7 \pm 4.2$ | $20.7 \pm 3.6$ | 0 | N/A | $27.4 \pm 5.9$ | $28.9 \pm 5.5$ | $1.5 \pm 5.9$ | **<0.046** |
| ECW–ICW ratio (%) | $75.4 \pm 11.3$ | $74.2 \pm 8.8$ | $-1.2 \pm 12.6$ | 0.254 | $70.4 \pm 11.5$ | $65.7 \pm 8.7$ | $-4.7 \pm 12.6$ | **0.002** |
| PhA (°) | $5.7 \pm 0.6$ | $5.8 \pm 0.6$ | $0.1 \pm 0.6$ | **0.031** | $6.2 \pm 0.9$ | $6.6 \pm 0.9$ | $0.4 \pm 0.8$ | **<0.001** |

BM: body mass. BMI: body mass index. BFM: body fat mass. BH: body height. FFM: fat-free mass. TBW: total body water. ECW: extracellular water. ICW: intracellular water. PhA: whole body phase angle. Mean Diff.: Mean difference. N/A: not applicable. A paired *t*-test was used. [†] Adjusted for BH. Statistically significant values are written in bold.

In terms of BMI classification, all categories improved in BM and BFM. First, participants in the normal BMI class had a significantly increased FFM, and in pre-obesity class they had a preserved FFM. However, participants in the obesity BMI class significantly lost FFM ($-1.5 \pm 3.6$ kg, $p = 0.032$), which represents 20% of their total BM loss ($-7.5 \pm 6.1$ kg).

In regard to BMI and PhA, the participants in the normal BMI class had a significantly increased average PhA ($5.8 \pm 0.6°$ and $6.0 \pm 0.6°$, $p = 0.03$), those in the pre-obesity BMI class had a non-significantly increased PhA ($6.0 \pm 0.8°$ and $6.1 \pm 0.9°$, $p = 0.146$), and those in the obesity BMI class had a significantly increased PhA ($6.1 \pm 0.8°$ and $6.5 \pm 1.0°$, $p = 0.015$).

In addition, the participants who lost BM < 5 kg had a significantly increased FFM ($0.8 \pm 3.2$ kg of FFM (57%) within $-1.4 \pm 1.8$ kg of BM loss), whereas the participants who lost BM $\geq$ 5 kg had a decreased FFM ($-2.2 \pm 3.9$ kg of FFM ($-25$%) within $-8.8 \pm 5.2$ kg of BM loss). The complete BM, BFM and FFM changes according to the BMI classification and BM loss for the whole sample are presented in Table 3.

**Table 3.** Body mass, BFM and FFM change according to BMI classification and BM loss.

| | BM (kg) | | | | BFM (kg) | | | | FFM (kg) | | | |
|---|---|---|---|---|---|---|---|---|---|---|---|---|
| | Baseline | FU | Mean Diff. | *p*-Value | Baseline | FU | Mean Diff. | *p*-Value | Baseline | FU | Mean Diff. | *p*-Value |
| BMI classification * | | | | | | | | | | | | |
| Normal BMI class [†] | $64.2 \pm 8.2$ | $63.0 \pm 8.2$ | $-1.2 \pm 2.2$ | **<0.001** | $14.7 \pm 4.2$ | $12.3 \pm 4.1$ | $-2.4 \pm 1.6$ | **<0.001** | $50.5 \pm 8.6$ | $51.4 \pm 9.1$ | $0.9 \pm 2.7$ | **<0.001** |
| Pre-obesity BMI class [††] | $80.7 \pm 9.8$ | $76.8 \pm 9.5$ | $-3.9 \pm 3.7$ | **<0.001** | $21.4 \pm 5.6$ | $17.8 \pm 5.9$ | $-3.6 \pm 2.7$ | **<0.001** | $57.7 \pm 11.3$ | $57.4 \pm 11.5$ | $-0.3 \pm 4.5$ | 0.613 |
| Obesity BMI class [†††] | $101.0 \pm 18.7$ | $93.5 \pm 18.6$ | $-7.5 \pm 6.1$ | **<0.001** | $34.3 \pm 10.6$ | $28.6 \pm 11.0$ | $-5.6 \pm 5.4$ | **<0.001** | $65.5 \pm 13.9$ | $64.0 \pm 12.9$ | $-1.5 \pm 3.6$ | **0.032** |
| Sub-analysis | | | | | | | | | | | | |
| BM loss < 5 kg | $71.3 \pm 14.4$ | $69.9 \pm 13.9$ | $-1.4 \pm 1.8$ | **<0.001** | $17.9 \pm 7.4$ | $15.5 \pm 7.4$ | $-2.3 \pm 1.4$ | **<0.001** | $53.5 \pm 10.7$ | $54.3 \pm 10.9$ | $0.8 \pm 3.2$ | **0.001** |
| BM loss $\geq$ 5 kg | $88.2 \pm 19.0$ | $79.4 \pm 17.3$ | $-8.8 \pm 5.2$ | **<0.001** | $26.5 \pm 10.9$ | $19.8 \pm 10.4$ | $-6.7 \pm 4.6$ | **<0.001** | $60.8 \pm 13.4$ | $58.6 \pm 12.6$ | $-2.2 \pm 3.9$ | **<0.001** |

* BMI classification by the World Health Organization [55]. [†] BMI 18.5–24.9 kg/m$^2$ ($N = 111$). [††] BMI 25–29.9 kg/m$^2$ ($N = 68$). [†††] BMI >30 kg/m$^2$ ($N = 38$). BM loss < 5 kg ($N = 165$), BM loss $\geq$ 5 kg ($N = 44$), increased BM ($N = 8$). Mean Diff.: Mean difference. A paired *t*-test was used. Statistically significant values are written in bold.

In regard to the correlation between BMI, BFM, FFM, and PhA, we reported the results in Figures S1–S3. In brief, our results showed a low negative association between BMI and PhA ($r = -0.22$, $p = 0.001$) and BFM and PhA ($r = -0.28$, $p < 0.001$) and a very low positive association between FFM and PhA ($r = 0.02$, $p = 0.806$), which was not statistically significant.

## 4. Discussion

### 4.1. Main Findings

In recent years, research on vegan diets has been growing. However, the main concern besides nutrient adequacy [65–68] is muscle mass loss during BM loss, probably due to lower energy/protein intake and/or due to a failure to introduce adequate PA (e.g., resistance workout) in combination with the dietary intervention. However, our study, for the first time, provides answers to some questions regarding BC changes during BM loss and the use of a vegan diet.

The main findings of our study can be divided into three main segments: (i) the vector of change of BMI, BFM per BH, FFM per BH, and PhA from initial values to FU, (ii) the examination of FFM change within the total BM loss according to BMI classification and (iii) the critical concern about FFM loss in the obese BMI class.

Our study showed that a WFPB lifestyle enabled desirable BC changes for the whole sample and for both women and men. Second, the participants who lost BM < 5 kg had a significantly increased FFM within their BM loss. However, the obesity BMI class significantly lost FFM (20% within the total BM loss). Third, this BMI class deserves a solution to this problem (e.g., increased protein intake at the start compared to the non-obese BMI class and/or a customized resistance workout) for further improvement of muscle mass preservation, especially since general resistance workouts at the start of the change is not safe or recommended for the obese BMI state.

### 4.2. Anthropometrics and BC Indices

Our results showed that participants (whole sample) were on average at the limit of pre-obesity BMI class state (25.4 kg/m$^2$). Furthermore, the WFPB lifestyle programme improved their BC status, namely, it decreased their BFM, preserved FFM and increased PhA. However, there are several specifics that need to be addressed. First, regardless of whether the women were within the normal BMI class (24.5 kg/m$^2$) but in the trend of increased BF % (28.4%); however, both women and men, on average, significantly preserved their FFM during BM loss ($-2.4$ and $-3.9$ kg). In addition, the most important findings may be that the participants who lost BM < 5 kg were able to preserve their FFM (actually they increased it), whereas the participants who lost BM $\geq$ 5 kg were not able to completely preserve their FFM; they lost $-2.2$ kg of FFM (25%) of all lost BM ($-8.8$ kg).

Importantly, this is 48 to 68% less that found in three randomized controlled studies using a WFPB diet for 16 weeks, where on average, obese participants (measured by DXA) lost from 37% to 42% of their lean mass within their total BM loss (6–7 kg) [27–30]. Of note, at this time, the research examining the effect of a vegan diet on BC change during BM loss was virtually non-existent or limited to one research group led by the respected Physicians Committee for Responsible Medicine. Although the following findings may not be directly transferrable to our general population, a 6-week-long case study was recently published that reported BC changes in a male professional powerlifter on a vegan diet. The results suggested that a combination of a vegan diet and resistance workouts may support the loss of BM and BFM while preserve and even increase FFM [69].

In our previous intervention control study from 2017 using a WFPB lifestyle for 10 weeks for overweight participants (measured by BIA), they lost only $-0.3$ kg (5%) of FFM within the total BM loss ($-5.6$ kg), whereas the obese subsample lost only $-0.9$ kg (12%) of FFM within the total BM loss ($-7.3$ kg) [53]. In addition, we also confirmed the positive impact of the WFPB lifestyle in a 10-week single-arm intervention study (with the continuation of some participants until the end of 36 weeks), where on average, overweight

participants (measured by BIA) lost only −0.1 kg (4%) of FFM within their total BM loss (−2.6 kg), whereas 18 participants who continued to follow the WFPB lifestyle programme for the next 26 weeks did not lose FFM at all within an additional 3 kg of BM loss [51]. The more favourable results of FFM preservation in our previous studies compared with our retrospective data analysis were most likely due to the study involvement of the participants and possibly a more extensive FU support.

Regardless, why are the obtained results for the obese BMI class, despite modest FFM loss, seen as excellent achievements? First, in regard to the obese BMI class and BM loss, in the first BM phase, it is generally recommended that the individual lose up to 5% of BM with diet and limited aerobic PA (e.g., slow walking) only, so this population is limited to performing resistance training that effectively preserves muscle mass during their BM loss. Second, specific to the WFPB diet, it is generally suggested they eat each well-designed meal ad libitum [27–30,51,53,59,70]; however, some participants may initially be "afraid" to eat until full satiety to ingest enough energy/protein. The reason for this is that people have heard all of their lives that when embarking on a BM loss diet they should eat less. Nevertheless, as suggested by several researchers, for an even better muscle preservation effect among obese BMI classes, this would most likely be achieved with a higher protein intake (at least in the first phase) and customized resistance training (e.g., the use of exercise machines at the gym) [17,31].

With regard to the PhA parameter, there is a lack of studies on vegans or when using vegan diet interventions. Nevertheless, one cross-sectional study examined PhA in 1013 healthy adults on a Mediterranean diet, which is known to be higher in plant-based origin food sources than typical for healthy adults on an omnivorous diet. The average PhA for women and men on a Mediterranean diet was higher than that for participants on an omnivorous diet and was found to be $5.6 \pm 0.7°$ (relative to $5.8 \pm 0.6°$ in our study after FU on a WFPB lifestyle programme) and $6.1 \pm 0.8°$ (relative to $6.6 \pm 0.9°$ in our study after FU on a WFPB lifestyle programme) but as expected it was the lowest among those with a higher BMI and an older age [71]. However, our results showed that after the FU, the obesity BMI class had the highest PhA (6.5°), followed by the pre-obesity BMI class (6.1°) and the normal BMI class (6.0°). Possible reasons may be related to the influence of different proportions of women and men on the individual BMI classification (e.g., men had a higher PhA than women) and a higher BMI status among men (e.g., average BMI in men was 27.1 kg/m$^2$ vs. 25.5 kg/m$^2$ in women). Since all BMI classification groups improved, we suggest that this improvement in PhA was related to better nutrition and PA (e.g., resistance workouts).

Having said that, we conclude with four important findings and one concern. First, in a broader context, the current Slovenian BC status is extremely worrying due to the high BFM of adults and older adults (measured by BIA) of both genders [72]. Second, obesity is defined as the accumulation of excess body fat and not simply excess BM, and this is very important also for individuals who are within the normal BMI class (e.g., have so-called sarcopenic obesity) [37,73]. Sarcopenic obesity is currently a major public health problem with increasing prevalence worldwide (e.g., up to 42% of adults) [74,75]. Third, an appropriate BM, BFM and FFM were found to be better controlled with a combination of dietary changes and PA [15–18], as we have found in this study and confirmed in our previous studies on "healthy adults", which included a support system for sustainable behaviour changes [51,53,76]. Last, the improvement in BC status requires appropriate types and frequencies of PA, which has further beneficial impacts on body image satisfaction that was further found to be associated with long-term BM management [77–79]. Regarding the concern, there is an open research question about health benefits that go beyond the visual body image, e.g., if a person with BMI in the normal range (24.5 kg/m$^2$) loses additional BM (which brings BMI to 22.5 kg/m$^2$) or if a person (e.g., female) with BF % in the normal range (25%) loses additional BF % (which brings BF % to 20%).

*4.3. Baseline Western-Type Lifestyle of the Participants and the Changed WFPB Lifestyle Programme*

Baseline evaluated dietary patterns of our participant were very much in line with a typical Western-type diet and lifestyle where there are many areas of possible improvement. Furthermore, this modern way of eating is characterized by excessive consumption of refined sugar, added fats, saturated fats, and dietary cholesterol on the one hand and insufficient intake of whole grains, legumes, fruits, vegetables and nuts on the other. This disparity plays a crucial role in the increasing incidence of obesity, type 2 diabetes and different forms of cardiovascular diseases [80,81].

The majority of our participants (181 (83%)) before joining our programme were either inactive or used slow walking (i.e., slow walking is not characterized as moderate-intensity PA that determines the recommended weekly PA) as their only PA; however, 48 (22%) participants used resistance workouts regularly (e.g., 8 (17%) of them used fitness studios where the rest used their own BM). Regardless, on average, we found that the biggest issue related to PA was inconsistency. An important issue here is that a low level of PA was found to be the 10th leading risk factor for morbidity [82], whereas a combination of being less physically active in combination with everyday prolonged sitting was found to be associated with an up to a 94% increase in all-cause mortality in women and 48% in men compared to individuals who reported the least amount of daily sitting and being the most physically active [83]. In this context, data for Slovenia for 2016 reported that only 56% of adults were physically active according to the definition (e.g., according to the old WHO recommendation); however, only 47.1% and 38.7% of men and women performed the recommended vigorous-intensity PA, and 32.9% and 29.9% of men and women achieved the recommended moderate-intensity PA, respectively [84].

Our participants joined the WFPB lifestyle programme that we briefly explained in the methods section but examined the dietary intake and lifestyle characteristics in our previous studies [59]. In brief, in our recent cross-sectional study on 151 active adult participants who joined our ongoing, community-based WFPB lifestyle optimization programme, measured via 3-day weighted dietary records, we showed that the WFPB dietary pattern was fully nutritionally adequate. The average energy intake in both sexes combined was 2057 kcal/d, while the macronutrient ratio was 58% carbohydrates (287 g/d), 7% fibres (70 g/d), 20% fat (44 g/d) and 15% protein (77 g/d) [59]. In addition, participants reported on average a moderate transport time (41.7 min/d), relatively low weekly and weekend prolonged daily sitting (4 and 5 h/d), good sleep quality (score 2.7 on Pittsburgh Sleep Quality Index), low stress (score of 0.29 on Perceived Stress Questionnaire) and relatively high amounts of total PA (5541.2 MET min/w). As suggested within the WFPB lifestyle programme, we found that the majority of participants implemented resistance workouts 2.7 times/w for at least 30 min at a time [76].

In conclusion, the PA recommended in our WFPB lifestyle programme, although more concrete, are in line with the newest WHO guideline recommendation stating that adults should undertake at least 150–300 min/week of vigorous-intensity PA or some equivalent combination of moderate-intensity and vigorous-intensity aerobic PA [85].

## 5. Strength, Limitations and Future Direction

The main strength of our study was that we included all data within the criteria (e.g., all "healthy adults" who had at least two successive measurements and joined the WFPB lifestyle programme), without BMI limitations or a previous preference for a vegan diet. In addition, among those, no one was excluded from the data analysis. Importantly, our analysed data were not used in our previous studies [51–53,76], thus avoiding duplication of data. In addition, our results were collected from adults non-intentionally through the spontaneous process of our ongoing, community-based WFPB lifestyle programme; thus, we automatically avoided the well-known bias effect of individuals being involved in research [86]. Of note, the analysed data were from participants from a wide geographical area in Slovenia, thereby excellently presenting a real (free-living) situation in regard to the

effects of the WFPB lifestyle programme on FFM preservation. Importantly, participants were not paid to participate in the programme. Furthermore, our results are in line with our previous results [51–53,76]. Finally, this is the first study that proactively addressed the issue of FFM preservation during BM and/or BFM loss using a vegan diet.

The main limitation of our study is its study design (i.e., it is a retrospective study) and the selection bias of the participants since the subjects ultimately agreed to join the WFPB lifestyle programme themselves after the introduction and consultation phases, which may contribute to their motivation/adherence. A significant contribution to adherence is also due to the supplemented WFPB diet and the support system where we followed up participants with our comprehensive, ongoing, community-based support system, described here and elsewhere [59,87]. However, in this study, the adherence to the WFPB lifestyle programme was not assessed.

For future studies, preferably randomized controlled trials using a well-designed vegan diet, we propose to regularly report BC changes. Moreover, we suggest replicating our results from this retrospective study and from previous intervention studies [51,53] and using the WFPB lifestyle for better FFM preservation during BM loss compared to using a well-designed vegan diet only. Of note, we understand that many studies are examining other outcomes and are interested solely in the effect of a vegan diet. However, our opinion is that ignorance of favourable changes using a vegan diet only by evaluating BMI while simultaneously losing a high proportion of muscle mass within the total BM loss is in most cases unacceptable. In this regard, the main concern we see is based on the fact that by preserving muscle mass during BM loss, we can expect a further loss of excess BM and/or BFM, which may improve the visual body image (with less excess body skin left behind) and consequently better adherence to the WFPB diet.

## 6. Conclusions

The results of our study suggest that a WFPB lifestyle that includes a supplemented WFPB diet, PA and support system effectively preserved FFM during BW and/or BF loss among healthy adults. In addition, the amount of specific PA (e.g., resistance workouts) that was prescribed was 2–3 times weekly for at least 30 min at a time and was executed mostly with only one's own BW. This amount and the type of PA are in line with recent WHO recommendations [85] and therefore are viable options for the general adult population. Moreover, we believe that preserving muscle mass represents the potential for further BM and/or BF loss, allowing individuals to improve both their visual body image and their adherence to a vegan diet.

Furthermore, the obesity BMI class still needs further improvement when implementing a WFPB lifestyle programme due to the nature of the obesity BMI class, where studies suggest a slightly higher intake, in the case of a vegan diet, of plant protein compared to the non-obese BMI class and/or a customized resistance training (e.g., the use of fitness machines). Regardless, our proposed paradigm on muscle mass preservation must be further verified in randomized control studies using a well-designed WFPB/vegan diet but with the addition of lifestyle components (e.g., resistance workout).

**Supplementary Materials:** The following supporting information can be downloaded at: https://www.mdpi.com/article/10.3390/obesities2020014/s1, Table S1: Anthropometrics and BC indices (mean ± SD). Figure S1: Association between PhA and BMI. Figure S2: Association between PhA and BFM. Figure S3: Association between PhA and FFM.

**Author Contributions:** Conceptualization, B.J. (Boštjan Jakše) and S.P.; methodology, B.J. (Boštjan Jakše) and U.G.; formal analysis, U.G.; investigation, B.J. (Boštjan Jakše), B.J. (Barbara Jakše) and S.P.; resources, B.J. (Barbara Jakše); writing—original draft preparation, B.J. (Boštjan Jakše); writing—review and editing, B.J. (Boštjan Jakše), B.J. (Barbara Jakše), S.P. and U.G.; supervision, S.P. and U.G.; project administration, B.J. (Barbara Jakše); funding acquisition, B.J. (Barbara Jakše). All authors have read and agreed to the published version of the manuscript.

**Funding:** The work did not need external funding.

**Institutional Review Board Statement:** This retrospective study was approved on 12 April 2021 by the Ethical Committee in the field of sport in Slovenia (No. 033–7/2021–2), while the trial was registered on 29 April 2021, at https://clinicaltrials.gov with number NCT04849689.

**Informed Consent Statement:** Not applicable.

**Data Availability Statement:** The data used to support the findings of this study are included within the article.

**Acknowledgments:** The authors wish to thank all participants who joined the WFPB lifestyle program. This work would not have been possible without them.

**Conflicts of Interest:** B.J. (Boštjan Jakše) and B.J. (Barbara Jakše) created the WFPB lifestyle program and S.P. is an active contributor. Part of the supplemented WFPB diet uses Herbalife Nutrition products from which B.J. (Boštjan Jakše), B.J. (Barbara Jakše) and S.P. receive royalty compensation. U.G. declares no conflict of interests related to this manuscript.

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
