# Peer review of "Ongoing Community-Based Whole-Food, Plant-Based Lifestyle Effectively Preserves Muscle Mass during Body Mass Loss"

_2673-4168, doi:10.3390/obesities2020014_

Round 1

Reviewer 1 Report

This paper presents an interesting investigation on changes in body composition assessed by bioelectrical impedance assessed through repeated measurements in a sample of 217 Slovenian adults who joined the WFPB community-based lifestyle program from 2016 to 2021. The topic covered is of scientific interest, the article is well written and clear. The major merit of the study is to examine the changes in body composition, considering the issue of FFM preservation during BM loss with the use of a vegan diet.

Having said that, the major problem hinges on considering the data for males and females together in the article. As is well known there are significant differences in the anthropometric measurements (from height to body composition parameters) of the two sexes and the higher frequency of women in the sample (as the authors themselves admit in the lines 383-386) has certainly influenced the averages of the total sample. You also reported in the supplementary (only) the significant differences between sexes. I, therefore, suggest reporting all results separately by sex. It is certainly very interesting to see if there have been different responses to the program in the two sexes.

Finally, here are my last minor remarks:

- page 2: I suggest strengthening the introduction concerning the relationships between lifestyle and body composition with recent publications. See for example: Maitiniyazi et al. https://doi.org/10.2147/DMSO.S325115 ; Zaccagni et al. doi: 10.23736/S0022-4707.17.07871-9

- page 3, line 391: Since we are referring to biological sex here, change "genders" to "sexes".

Reviewer 2 Report

This is an observational study on the effect of a WFBP life style intervention on changes in bw and bc. The authors provide evidence that the WFBP program was capable to maintain or even preserve most of FFM with weight loss.

My concerns are as follows,

there is no control group, the authors did not present data on patients who refused to participate or have withdrawn from the program. There is n ITT analysis. In addition, there is need of a power analysis.

In weight loss programs the variance in adherence has a major impact on outcome, i.e. , weight loss (see AJCN 2007;85:346-354). To address that issue the authors are asked to first use Kevin Hall's model to predict individual weight changes (see Lancet 2011; 378(9793):826-37). Then, the predicted weight loss should be compared with the measured weight loss according to Am J Physiol Endocrinol Metab. 2012 ;302:E441-448. Finally, the variance in adherence should be compared with the variances in changes in bw, FM and FFM. This would allow an evaluation of the intervention program.

The authors have uncritically used the read out of their BIA device. Since BIA mostly estimates resistance related to arms and legs while the contribution of the trunk to whole body resistance is minor only, BIA cannot be used to assess VAT. In the Tanita algorithm (which is unknown to the authors) VAT is indirectly calculated based on statistical associations between individual body components. Thus, these data should be skipped from the results. The same comes true for BMC which cannot be measured by BIA. In addition, some of the read outs are redundant, e.g., %FM is biased by bw and thus should be skipped from the data presentation. For inter-individual comparisons height2-adjusted data of FM and FFM should be given. Finally, the precision and the MDC in terms of FM of BIA have to be taken into account: Which is the degree of change in FM which can be assessed with confidence?

Round 2

Reviewer 1 Report

The manuscript is ready for publication. I suggest only one last small change to the caption of Table 2: "by gender" should be replaced by "by sex". 

Author Response

Dear, 

thank you very much. 

We have made the suggested change. 

With respect, 

Boštjan Jakše, PhD

Reviewer 2 Report

Sorry, the authors did not get the point. They should also be open for the idea that a reviewer wants to help them rather than to criticize them.

There are two points left to address.

First, since in any weight loss program adherence is the major determinant of outcome it has to be addressed according to the suggestions given in my first review. This is because the outcome of this program cannot be evaluated at moderate or low adherence. It is simply not enough to describe the effect of a diet without taking into account the variance in adherence. In its present form the authors can only speculate that the observed weight loss is partly due to the intervention and partly due to the adherence to that intervention. From a scientific point of view this conclusion has limited value.

Second, the use of BIA in weight loosing patients is still a issue to be carefully addressed. If the authors have additional DXA data obtained in the same subjects they may took another look at that data. Simply referring to data which they do not show has no scientific value. In addition, DXA is a reference rather than a gold standard method. Thus, DXA also has problems addressing the composition of weight changes with confidence. To have a robust reference the 4C-model has to be used.  

Author Response

Dear, 

To the limitations, we added a sentence related to the adherence of this mechanistic study. Other topics (e.g., 4C model) and whether DEXA method is a reference or a gold standard were not further discussed (we believe it is beyond the scope of our study). 

In addition, when we mentioned that our BIA model was validated against the DEXA method (from other researchers), we were thinking exclusively of our model used (Tanita 780 S MA) and not tha we have duplicated data (also from the DEXA) method). 

In addition, we also added the conclusion section that somehow "disappeared" in the prevoious version of the manuscript - we apologize. 

With respect, 

Boštjan Jakše, PhD
